# Hybrid Active-and-Passive Relaying Model for 6G-IoT Greencom Networks with SWIPT

**DOI:** 10.3390/s21186013

**Published:** 2021-09-08

**Authors:** Sumit Gautam, Sourabh Solanki, Shree Krishna Sharma, Symeon Chatzinotas, Björn Ottersten

**Affiliations:** 1Interdisciplinary Centre for Security, Reliability and Trust (SnT), University of Luxembourg, L-1855 Luxembourg, Luxembourg; shree.sharma@uni.lu (S.K.S.); symeon.chatzinotas@uni.lu (S.C.); bjorn.ottersten@uni.lu (B.O.); 2Huawei Technologies Sweden AB, Vädursgatan 5, 412 50 Göteborg, Sweden

**Keywords:** 5G and beyond/6G wireless networks, Greencom, IoT, passive repeater, relaying systems, SWIPT

## Abstract

In order to support a massive number of resource-constrained Internet-of-Things (IoT) devices and machine-type devices, it is crucial to design a future beyond 5G/6G wireless networks in an energy-efficient manner while incorporating suitable network coverage expansion methodologies. To this end, this paper proposes a novel two-hop hybrid active-and-passive relaying scheme to facilitate simultaneous wireless information and power transfer (SWIPT) considering both time-switching (TS) and power-splitting (PS) receiver architectures, while dynamically modelling the involved dual-hop time-period (TP) metric. An optimization problem is formulated to jointly optimize the throughput, harvested energy, and transmit power of a SWIPT-enabled system with the proposed hybrid scheme. In this regard, we provide two distinct ways to obtain suitable solutions based on the Lagrange dual technique and Dinkelbach method assisted convex programming, respectively, where both the approaches yield an appreciable solution within polynomial computational time. The experimental results are obtained by directly solving the primal problem using a non-linear optimizer. Our numerical results in terms of weighted utility function show the superior performance of the proposed hybrid scheme over passive repeater-only and active relay-only schemes, while also depicting their individual performance benefits over the corresponding benchmark SWIPT systems with the fixed-TP.

## 1. Introduction

### 1.1. General Motivation

The world witnesses the introduction of a novel generation of wireless communications approximately every ten years. With the current speedy deployment of hardware for the fifth-generation (5G) of mobile wireless systems in 2021, we are now at a juncture to anticipate what lies ahead for the constitution of the sixth generation (6G) mobile cellular systems [1]. The incoming crucial updates to the hardware equipment for 5G may require dense infrastructure deployment [2]. In future, it will become important to leverage the 5G architecture for internet-of-things (IoT) systems, along with the introduction of some methods to increase the coverage area as well as considering energy harvesting (EH) frameworks while embracing the concept of Green Communications (Greencom)-IoT systems.

### 1.2. Background to the Considered Topics

With the wide deployment of 5G systems, it is becoming crucial to understand the main notion behind the working of such technology. The 5G systems may be implemented in low-band, mid-band or high-band millimeter-wave (mmWave from 24 GHz up to 100 GHz are expected to be used for 5G) [3]. The air interface defined by the 3rd Generation Partnership Project (3GPP) for 5G is known as New Radio (NR), and the specification is subdivided into two frequency bands, frequency range-1 (FR1) (below 6 GHz) and frequency range-2 (FR2) (in the mmWave regime) [4], each with different capabilities [5]. Considering the low-band, 5G uses a frequency range similar to 4G cellphones, i.e., 600–850 MHz. However, the 5G is able to provide download speeds little higher than 4G, i.e., 30–250 megabits per second (Mbit/s) [6]. The range and coverage area of the low-band 5G cell towers area similar to the ones in 4G. Regarding the mid-band 5G, microwaves of 2.5–3.7 GHz are used which allow speeds of 100–900 Mbit/s, with each cell tower providing service up to several kilometers in radius. This level of service is the most widely deployed within many metropolitan areas in 2020. The high-band 5G uses frequencies of 25–39 GHz, near the bottom of the mmWave band. In this vein, there is also a possibility that higher frequencies will be used in the future, e.g., for the 6G systems. Using this scheme often achieves download speeds in the gigabit-per-second (Gbit/s) range, which is comparable to cable internet. However, mmWave bands have a more limited range, requiring many small cells [7].

Since the proposed 5G setting will be more prominent in the coming future, it cannot be denied that the future 6G systems will leverage a lot of the hardware architectures and techniques from the 5G framework. In this regard, deployment of several small-cells in various places on the planet will become cumbersome and hence some supporting techniques need to be sought to enhance the network coverage. Cooperative communications, which works on the principle of introducing repeater(s) or relay(s) between the transmit source and end-user, has been established as a promising technique to address the limited coverage concerns. Herein, repeater(s) or relay(s) assist in forwarding the transmit signal towards the end-user by preserving the signal properties and power, which can enable enhanced coverage. In this vein, the cooperative mechanism may be performed via single or multiple signal hops [8,9].

As the technological advancements in the field of wireless communications continue to amaze the mankind in several ways, a major hurdle in this progress is our dependence on the energy sources, which are as strong as on the gadgets themselves [10]. An early EH concept of getting energy from/through the air to power all equipment, was proposed by Nikola Tesla in the late 1890s to early 1900s [11,12]. Ever since then, this concept has found its applications in several fields of research including vehicles [13], wireless gadgets [14], flying objects [15], etc. More recently, several researchers have proposed using RF signals for facilitating simultaneous wireless transmission of information and energy , which involves the traditional information receiver unit along with an EH module. This idea of joint information and energy transmission is likely to play significant role in emerging technologies [16,17,18]. Applications like wearable devices, wireless sensor nodes etc., are expected to adopt the proposed wireless recharging alternative more rapidly. With the increasing demand for energy to operate the devices, it has almost become a necessity to adopt such techniques to compensate for rapidly draining battery sources [19,20,21].

In order to investigate the performance above-mentioned scenarios, optimization of system parameters such as, e.g., data throughput, harvested energy, or transmit power, come in handy. Herein, the mathematical formulation of the optimization problem corresponding to the wireless scenario is considered under a certain set of constraints. In case the formulated optimization problem is convex, it can be solved directly via simple methods, such as gradient descent algorithm [22]. For the deterministic and continuous constrained optimization under non-linear programming, obtaining a direct solution is cumbersome [23]. Herein, suitable transformations or approximations may be employed to convexify the primal problem in order to obtain a suitable solution [24,25]. Alternatively, a Lagrange dual method based solution could be a possibility, which operates on the dual regime corresponding to the main problem. It is noteworthy that the dual problem is always convex and hence a suitable solution can be sought [22,26]. However, in this case, it is important to check the duality gap between the primal solution and the one obtained via dual method. Extremely less or zero duality gap indicates that the optimal solution has been achieved, while a large duality gap indicates a sub-optimal solution is obtained [27]. On the other hand, an optimization problem with fractional objective is extremely challenging to solve. In this context, Dinkelback algorithm based solution is found to be most effective [28,29]. In the context of this paper, we leverage some steps from the above-mentioned techniques to solve the formulated problem, which will be presented in the later sections. To facilitate the reading process, we have summarized the list of abbreviations in Table 1. In the following, we discuss some related works to the framework considered in this paper.

### 1.3. Related Works

To enhance the network coverage of wireless communications systems, certain well-known methods such as passive repeater (PR) and/or relaying have shown immense potential [30]. A PR is a battery-less device used to reflect the transmit signal (with encoded data) towards the desired user, where it may be utilized to energize the IoT devices as well as to enable the deployment of battery-free devices in future networks [31]. Noticeably, traditional backscattering may be considered as a special form of PR. In case of an active relay (AR), the transmitted signal from the source is received at the relay device and re-transmitted to the end-user after either a direct amplification according to the amplify-and-forward (AF) protocol, or a decoding or quantization process before its re-transmission as a re-encoded signal with message suppression, according to the decode-and-forward (DF) protocol [32,33]. Regarding the facilitation of EH at the end-user, the simultaneous wireless information and power transfer (SWIPT) technique seems promising [34,35]. In this regard, the power-splitting (PS) [36] and time-switching (TS) [37] schemes have been widely considered to deploy SWIPT receivers in practice.

Considering the latest trends in relaying methodologies pertaining to the recent advancements in electromagnetic materials [38,39,40,41], the notion of Reconfigurable Intelligent Surface (RIS) [42] or Intelligent Reflecting Surface (IRS) [43] holds immense potential. Concretely, an RIS is a two-dimensional surface that interacts with the incoming electromagnetic waves using several small reflecting elements that are electronically tuned [44]. More specifically, a transmitted signal intended for an end-user may be more efficiently sent over a wireless medium, where the channel could now be controlled via an RIS such that a metal plate is rotated or bent to redirect the corresponding incident wave [45]. The meta-materials considered herein are expected to inherit some extra-ordinary electromagnetic features that are unnatural [46], such as, adverse refraction [47], flawless assimilation/absorption, and atypical reflection/scattering [48]. To ensure the possibilities discussed above, an ideal RIS must be comprised of large number of elements having controllable properties [49], wherein any alteration to the reflection coefficient (e.g., phase shift or re-modulation) of the elements can control the beamforming of the incident waves.

However, the above prospect of realizing RIS seems promising, there are still several challenges associated while considering its practical implementation aspect [50]. In this vein, three major concerns include: (i) Difficulty in providing continuous and uninterrupted power-supply to the RIS elements given their active nature; (ii) continuous switching methods to adapt the RIS elements according to the reflection needs, which require additional computational resources and energy; and (iii) management of thermal noise at the very small RIS elements (given their designs are more suitable for high frequency operations) becomes an issue, primarily due to continuous frictional operations and power supply. Besides, it is extremely difficult to ensure proper supply of energy where power resources are scarce. Therefore, it becomes natural to seek the PR alternatives as other complementary methods to the RIS. In this manner, the benefits of employing the reflectors may be harnessed using the passive means in the instances where the deployment of active-RIS becomes challenging [51].

Recent developments in the area of PR and AR type of schemes have focused on important aspects such as cooperation enhancement, however, with the consideration of a linear EH model [52,53]. In [54], the authors presented an interesting framework for wireless energy beamforming in the reflector-and-relay based communications system, wherein a non-linear EH mechanism is considered at the intermediary device but not at the end-user. Additionally, most related works focus on the maximization of either the signal-to-noise ratio (SNR) [55], or the throughput [56]. However, it is important to optimize the joint metrics of data throughput, harvested energy and transmit power of the SWIPT-enabled 6G systems. On one hand, it is noteworthy that the reflector-only mechanism may not be suitable for dense deployment scenarios [31,57]. Whereas on the other hand, the power-dependence of relays is critical in the relaying-only scenarios, wherein a power-failure may lead to the collapse of the entire wireless system without any backup. In this regard, the hybrid PR and AR based scheme may ensure a sustainable wireless ecosystem where both devices may supplement or complement each other accordingly. Moreover, in the context of the hybrid framework, a comparative study between the PS and TS SWIPT schemes with non-linear EH model along with the design of dynamic dual-hop time-period (TP) has not yet been investigated in the literature so far.

### 1.4. Our Contributions

In this invited paper, we present a dual-hop system wherein a transmit source transfers both information and energy to a SWIPT-enabled end-user employing either PS or TS architecture. We investigate three distinct Greencom systems incorporating a PR node, an AR node, and a novel hybrid active-and-passive relaying node, respectively; which either reflects and/or re-transmits the information-and-energy signal to the end-user, in addition to the weak direct link. The novelty and main contributions of this work are listed as follows.
(i)In contrast to the existing reflector-only and relaying-only techniques, we propose a novel hybrid active-and-passive relaying scheme to facilitate SWIPT to a PS- or TS-enabled end-user, along with the dynamic designing of dual-hop TP, under certain receive processing assumptions.(ii)We formulate a novel problem (incorporating the three systems) to maximize the weighted utility function comprising data throughput, harvested energy and transmit power, subjected to some quality-of-service (QoS) constraints. Unlike the other works that assume equal time-intervals in the two hops (also considered herein as the benchmarks), we present a framework to dynamically design the TP for the dual-hop link, along with the computation of the PS or TS factors.(iii)In order to solve the aforementioned problems, we present two distinct methods based on the Lagrange dual technique and Dinkelbach method assisted convex programming, respectively, where both the approaches yield an appreciable solution within polynomial computational-time.(iv)The effectiveness of the proposed hybrid active-and-passive relaying scheme is shown over the reflector-only and relay-only schemes for both PS and TS SWIPT schemes via numerical analysis, with individual benefits shown over their respective benchmark designs having a fixed TP.

### 1.5. Further Organization of the Paper

The remainder of this paper is organized as follows. Section 2 presents a description of the system model and Section 3 provides an analysis of the considered Greencom networks. Section 4 focuses on the joint optimization of data throughput, harvested energy and transmit power. Section 5 presents the simulation results. Finally, Section 6 concludes the paper.

## 2. System Model

We consider three wireless Greencom network scenarios for investigation, with a single transmit source, a single reflection and/or re-transmission device (i.e., either PR or AR or both), and a single user at the receiving end, as represented in Figure 1. The transmit source communicates with the end-user via two half-duplex communication links within a single time-slot. In particular, besides the conventional direct link, we assume the availability of a couple of intermediary links to assist in the delivery of the desired transmit signal. The end-user is assumed to be capable of performing both information decoding (ID) and energy harvesting (EH) simultaneously.

Let us define $P_{T}$ as the transmit power at the source, B as the overall bandwidth, and *T* as the TP for further analysis. In each system, the direct link is assumed to be active throughout the TP, with a pre-defined channel coefficient of $h_{1,D}$. For analytical convenience, we assume that the channel state information (CSI) is known to the intended devices. The transmit source emits the signal intended for the corresponding SWIPT-enabled end-user. The process is carried out using a direct link through the whole TP, with additional dual-hop assistance via a reflecting and/or re-transmitting device over the same TP. More specifically, the source transmits a symbol s∈C, which is received by the destination and the reflecting and/or re-transmitting device. The propagation path of the signal is sufficiently small due to the short communication distance. In this vein, the destination node is considered to utilize adequate delay compensation methods to synchronize and coherently process the received signals from the second hop and the direct link. Without loss of generality, we assume $E[|s{|}^{2}]=1$, with operation E[·] denoting the expectation value. The signal received by the end-user via direct link is defined as: $y_{1,D}=\sqrt{P_{T}}\;h_{1,D}\;s+n_{u_{a}}$, where $n_{u_{a}}$ is the additive white Gaussian noise (AWGN) at the end-user which is an independent and identically distributed (i.i.d.) complex Gaussian random variable with zero mean and variance $\sigma_{n_{u_{a}}}^{2}$.

Concerning the SWIPT-enabled receiver device, two different SWIPT schemes namely PS or TS, respectively, may be adopted. For the PS scheme, an optimal fraction of the received signal is provided to the information decoder and the remaining part is provided to the energy harvester. In this case, the PS ratio is denoted by ρ, where 0≤ρ≤1. Alternatively, when the receiver incorporates the TS scheme, we define a time switching ratio, τ, where 0≤τ≤1. In particular, for the first fraction of time period, all the received signal power is used for harvesting energy, whereas during the remaining time, information decoding from the received signal takes place.

In order to estimate the amount of harvested energy at the end-user, we adopt the sigmoidal function based non-linear EH model [27,58], defined as
(1)$$E\left(x\right)=\frac{E^{\prime }}{1-\phi }\cdot \left(\frac{1}{1+e^{\left(-\alpha x+\alpha \beta \right)}}-\phi \right),$$
where *x* is the input power at the energy harvesting module, $\phi \overset{\Delta }{=}\frac{1}{1+\exp(\alpha \beta )}$, the constant $E^{\prime }$ is obtained by determining the maximum harvested energy on the saturation of the energy harvesting circuit, and α and β are specific to the capacitor and diode turn-on voltage metrics at the EH circuit. Practically, a standard curve-fitting tool based on analytical data may be used to decide the appropriate values of $E^{\prime }$, α, and β. A comparison between the linear and non-linear EH models is depicted in Figure 2, where the energy conversion efficiency for linear EH module is assumed to be 0.75, $E^{\prime }$ = 2.8 mJ, α = 1500, and β = 0.0022 (for the non-linear EH module) [27,58]. The non-linearity introduced due to the diode and capacitor elements is observed at lower input power at the EH module while a constant EH operation is seen for higher values of input power which implies the saturation at diode element of the EH module. On the other hand, the linear EH model increases constantly based on the increasing input power at the EH module, without considering the saturation points of the diode element [58]. This is due to the assumption of a constant energy conversion efficiency at the EH module.

The harvested energy using the PS and TS schemes are respectively given by
(2)$$E^{\mathrm{PS}}\left(x,\rho \right)=\frac{E^{\prime }}{1-\phi }\cdot \left(\frac{1}{1+e^{\left(-\alpha \rho x+\alpha \beta \right)}}-\phi \right),$$
(3)$$E^{\mathrm{TS}}\left(x,\tau \right)=\tau \cdot \frac{E^{\prime }}{1-\phi }\cdot \left(\frac{1}{1+e^{\left(-\alpha x+\alpha \beta \right)}}-\phi \right).$$

The energy harvested at the end-user using the PS and TS schemes via direct link signal is given by $E_{1,D}^{\mathrm{PS}}\left(E[\right|y_{1,D}{|}^{2}],\rho )$ and $E_{1,D}^{\mathrm{TS}}\left(E[\right|y_{1,D}{|}^{2}],\tau )$, computed according to (Equation 2) and (Equation 3), respectively. Further, we assume that the normalized time slots use the terms power and energy interchangeably.

The effective signal-to-noise ratio (SNR) via direct link as seen at the ID branch of the end-user, respectively for PS and TS schemes, is given by
(4)$$\Upsilon_{1,D}^{\mathrm{PS}}=\frac{\left(1-\rho \right)\;P_{T}\;{\left|h_{1,D}\right|}^{2}}{\left(1-\rho \right)\;\sigma_{n_{u_{a}}}^{2}+\sigma_{n_{u_{ID}}}^{2}},\;\mathrm{and}\;\Upsilon_{1,D}^{\mathrm{TS}}=\frac{P_{T}\;{\left|h_{1,D}\right|}^{2}}{\sigma_{n_{u_{a}}}^{2}+\sigma_{n_{u_{ID}}}^{2}},$$
where $n_{u_{ID}}\in \mathrm{CN}\left(0,\;\sigma_{n_{u_{ID}}}^{2}\right)$ is the noise introduced by the baseband processing circuit. The effective throughput achieved at the end-user, combining the PS and TS schemes is given by
(5)$$R_{1,D}=\left\{\begin{array}{c} R_{1,D}^{\mathrm{PS}}=B\log_{2}\left(1+\Upsilon_{1,D}^{\mathrm{PS}}\right)\\ R_{1,D}^{\mathrm{TS}}=\left(1-\tau \right)B\log_{2}\left(1+\Upsilon_{1,D}^{\mathrm{TS}}\right) \end{array}\right.$$

## 3. Analysis of Greencom Network Scenarios

In this section, we present the discussion on the considered Greencom network scenarios in detail. To proceed, we define the time-splitting factor (TSF) of TP as δ, where 0<δ<1, to assist the dual-hop mechanism. It is evident that significant benefits may be obtained by employing dynamic TSF in contrast to the equal-time period-based splitting in dual-hop cooperative systems [59,60]. In this vein, it is noteworthy that the phenomenon of dynamic TSF modelling in the context of AF relaying primarily translates to a better power optimization and scheduling mechanism at the transmitter, while ensuring an improved management of the network resources. Specifically, we assume that the first-hop spans over δT of the TP while in the second hop, the remainder of the TP, i.e., (1−δ)T, is allocated for successful reflection and/or re-transmission of the intended signal from the PR and/or AR device(s) to the end-user. Additionally, we assume that the direct and indirect links are managed via different supervised signalling methods (e.g., use of separate directional antenna elements). More specifically, the transmission via direct link is assumed to be active for the whole time duration *T*, while the transmission through the indirect link remains active only for the stipulated δT duration. Following this assumption, it is needless to mention that the interference between the signal via (weaker) direct and (stronger) indirect links will be negligible, and thus this possible interference aspect is discarded in the further analysis.

### 3.1. Traditional Passive Repeater or Active Relay-Based Systems

In this case, we assume the availability of either reflection-antenna equipped PR device as the reflector node, or a single antenna based AR node as a re-transmission device, on the second hop. Further, we refer the two above-mentioned systems as PR device-based Greencom network (PGN) and AR device-based Greencom network (AGN), respectively. The channel coefficient in the first phase is defined as $h_{1}$.We represent the signal seen at the reflecting/re-transmitting device as: $r_{1}=\sqrt{P_{T}}\;h_{1}\;s+n_{1}$, where $n_{1}$ is the AWGN introduced due to the reflector/relay, which is an i.i.d. complex Gaussian random variable with zero mean and variance $\sigma_{n_{1}}^{2}$. In this context, we express $\left(r_{1},h_{1},n_{1}\right):=\left\{\left(r_{1,P},h_{1,P},n_{P}\right)\;:\;\mathrm{For}\;\mathrm{PGN};\left(r_{1,R},h_{1,R},n_{R}\right)\;:\;\mathrm{For}\;\mathrm{AGN}\right\}$.

The SNR and the effective data throughput expressions are respectively defined as
(6)$$\Upsilon_{1}=\frac{P_{T}\;{\left|h_{1}\right|}^{2}}{\sigma_{n_{1}}^{2}},\;\mathrm{and}\;R_{1}=\delta \;B\;\log_{2}\left(1+\Upsilon_{1}\right),$$
where $\left(\Upsilon_{1},R_{1},\sigma_{n_{1}}^{2}\right):=\{\left(\Upsilon_{1,P},R_{1,P},\sigma_{n_{P}}^{2}\right)\;:\;\mathrm{For}\;\mathrm{PGN};$$\left(\Upsilon_{1,R},R_{1,R},\sigma_{n_{R}}^{2}\right)$ : ForAGN}.

In the second hop, the channel coefficient is defined as $h_{2}$. In case of PGN, the intended signal from the PR is reflected to the end-user with the corresponding efficiency coefficient of $\sqrt{\eta }$, where $0<\sqrt{\eta }\leq 1$. Whereas in the case of AGN, the AR re-transmits the signal after scaling it by a complex amplification coefficient *w*. It is noteworthy that in the case of AF protocol, the amplification coefficient is a complex number which incorporates the amplitude as well as the direction/phase components. However, we only consider the constraint on amplitude component and ignore the directional/phase part in this work for simplifying the further analysis. In order to ensure feasibility of the AGN system, we impose an upper bound on the total relay power defined by: ${0<|w|}^{2}\leq P_{R}$, where $P_{R}=\frac{P^{⋆}-P_{T}}{P_{T}{\left|h_{1,R}\right|}^{2}+\sigma_{n_{R}}^{2}}$ is the maximum overall available power at the relay, with the transmitter-relay system bounded by an overall power of $P^{⋆}$, such that $P^{⋆}>\max\left(P_{T},P_{R}\right)$. To proceed, we define the following metric:(7)$$\Xi =\left\{\begin{array}{c} \;\sqrt{\eta }\;\;:\;{\mathrm{PR}}^{\prime }s\;\mathrm{Reflection}\;\mathrm{Coefficient}\;\mathrm{in}\;\mathrm{PGN}\\ \;w\;\;:\;{\mathrm{AR}}^{\prime }s\;\mathrm{Amplification}\;\mathrm{Coefficient}\;\mathrm{in}\;\mathrm{AGN} \end{array}\right.$$

The received signal as seen at the end-user via reflection or re-transmission from the PR or AR device, respectively, is represented as $y_{2}=\Xi \;h_{2}\;r_{1}+n_{u_{a}}$, where $\left(y_{2},h_{2},r_{1}\right):=\{\left(y_{2,P},h_{2,P},r_{1,P}\right)\;:\;\mathrm{For}\;\mathrm{PGN};$$\left(y_{2,R},h_{2,R},r_{1,R})\;:\;\mathrm{For}\;\mathrm{AGN}\right\}$.

We define the SNR obtained at the ID block of the end-user via second hop link, according to the PS and TS schemes, as
(8)$$\Upsilon_{2}^{\mathrm{PS}}=\frac{{\left|\Xi \right|}^{2}\;\left(1-\rho \right)\;P_{T}\;|h_{1}{|}^{2}\;{\left|h_{2}\right|}^{2}}{{\left(1-\rho \right)\;(|\Xi |}^{2}\;|h_{1}{|}^{2}\;\sigma_{n_{1}}^{2}+\sigma_{n_{u_{a}}}^{2})+\sigma_{n_{u_{ID}}}^{2}},$$
(9)$$\Upsilon_{2}^{\mathrm{TS}}=\frac{{\left|\Xi \right|}^{2}\;P_{T}\;|h_{1}{|}^{2}\;{\left|h_{2}\right|}^{2}}{{\left|\Xi \right|}^{2}\;{\left|h_{1}\right|}^{2}\;\sigma_{n_{1}}^{2}+\sigma_{n_{u_{a}}}^{2}+\sigma_{n_{u_{ID}}}^{2}},$$
where $\left(\Upsilon_{2}^{\mathrm{PS}},\Upsilon_{2}^{\mathrm{TS}},h_{1},h_{2},\sigma_{n_{1}}^{2}\right):=\{(\Upsilon_{2,P}^{\mathrm{PS}},\Upsilon_{2,P}^{\mathrm{TS}},h_{1,P},h_{2,P},$$\sigma_{n_{P}}^{2})\;:\;\mathrm{For}\;\mathrm{PGN};$$(\Upsilon_{2,R}^{\mathrm{PS}},\Upsilon_{2,R}^{\mathrm{TS}},$$h_{1,R},h_{2,R},$$\sigma_{n_{R}}^{2})$:ForAGN}. The effective throughput achieved at the end-user, incorporating the respective PS and TS schemes is defined as
(10)$$R_{2}=\left\{\begin{array}{c} R_{2}^{\mathrm{PS}}=\left(1-\delta \right)B\log_{2}\left(1+\Upsilon_{2}^{\mathrm{PS}}\right)\\ R_{2}^{\mathrm{TS}}=\left(1-\tau \right)\left(1-\delta \right)B\log_{2}\left(1+\Upsilon_{2}^{\mathrm{TS}}\right) \end{array}\right.,$$
where $\left(R_{2},R_{2}^{\mathrm{PS}},\Upsilon_{2}^{\mathrm{PS}},R_{2}^{\mathrm{TS}},\Upsilon_{2}^{\mathrm{TS}}\right):=\{(R_{2,P},R_{2,P}^{\mathrm{PS}},\Upsilon_{2,P}^{\mathrm{PS}},$$R_{2,P}^{\mathrm{TS}},$$\Upsilon_{2,P}^{\mathrm{TS}})\;:\;\mathrm{For}\;\mathrm{PGN};(R_{2,R},R_{2,R}^{\mathrm{PS}},$$\Upsilon_{2,R}^{\mathrm{PS}},$$R_{2,R}^{\mathrm{TS}},\Upsilon_{2,R}^{\mathrm{TS}})\;:\;\mathrm{For}$AGN}.

The PS and TS expressions for harvested energy are denoted as $E_{2}^{\mathrm{PS}}\left(E[\right|y_{2}{|}^{2}],\rho )$ and $E_{2}^{\mathrm{TS}}\left(E[\right|y_{2}{|}^{2}],\tau )$, computed according to (Equation 2) and (Equation 3), respectively, where $\left(E_{2}^{\mathrm{PS}},E_{2}^{\mathrm{TS}},y_{2}\right):=\{\left(E_{2,P}^{\mathrm{PS}},E_{2,P}^{\mathrm{TS}},y_{2,P}\right)\;:\;\mathrm{For}\;\mathrm{PGN};$$\left(E_{2,R}^{\mathrm{PS}},E_{2,R}^{\mathrm{TS}},y_{2,R})\;:\;\mathrm{For}\;\mathrm{AGN}\right\}$.

At the end-user, the overall data throughput achieved incorporating the yield from the second hop and direct link, for respective PS and TS schemes, is defined as
(11)$$R_{U}=\left\{\begin{array}{c} R_{U}^{\mathrm{PS}}=R_{1,D}^{\mathrm{PS}}+R_{2}^{\mathrm{PS}}\\ R_{U}^{\mathrm{TS}}=R_{1,D}^{\mathrm{TS}}+R_{2}^{\mathrm{TS}} \end{array}\right.,$$
where $\left(R_{U},R_{U}^{\mathrm{PS}},R_{U}^{\mathrm{TS}}\right):=\{\left(R_{U,P},R_{U,P}^{\mathrm{PS}},R_{U,P}^{\mathrm{TS}}\right)\;:\;\mathrm{For}\;\mathrm{PGN};$$\left(R_{U,R},R_{U,R}^{\mathrm{PS}},R_{U,R}^{\mathrm{TS}})\;:\;\mathrm{For}\;\mathrm{AGN}\right\}$. Note that due to the involvement of the time related metrics, the expressions in (Equation 10) are indicative of the overall data throughput and not the actual rate. Hence, it is possible to combine the data throughputs from the two links following a proper synchronization process at the receiver. This is not to be confused with the case of overall rate (traditionally measured in bits per seconds), wherein it is necessary to perform the SNR combining methods, such as, e.g., maximum ratio combining (MRC) [61].

The effective harvested energy expression incorporating both PS and TS schemes is:(12)$$E_{U}=\left\{\begin{array}{c} E_{U}^{\mathrm{PS}}=E_{1,D}^{\mathrm{PS}}\left(E[\right|y_{1,D}{|}^{2}],\rho )+\left(1-\delta \right)E_{2}^{\mathrm{PS}}\left(E[\right|y_{2}{|}^{2}],\rho )\\ E_{U}^{\mathrm{TS}}=E_{1,D}^{\mathrm{TS}}\left(E[\right|y_{1,D}{|}^{2}],\tau )+\left(1-\delta \right)E_{2}^{\mathrm{TS}}\left(E[\right|y_{2}{|}^{2}],\tau ) \end{array}\right.$$
where $\left(E_{U},E_{U}^{\mathrm{PS}},E_{U}^{\mathrm{TS}}\right):=\{\left(E_{U,P},E_{U,P}^{\mathrm{PS}},E_{U,P}^{\mathrm{TS}}\right)\;:\;\mathrm{For}\;\mathrm{PGN};$$\left(E_{U,R},E_{U,R}^{\mathrm{PS}},E_{U,R}^{\mathrm{TS}})\;:\;\mathrm{For}\;\mathrm{AGN}\right\}$.

### 3.2. Proposed Hybrid Active-and-Passive Relaying Scheme

We present herein the proposed hybrid active-and-passive (HAP) relaying scheme in Greencom networks to facilitate SWIPT to the end-user over the second hop. In this case, the HAP relaying node receives the transmit signal on both the PR and AR system equipped with single antenna each, over the above-defined channel coefficients of $h_{1,P}$ and $h_{1,R}$, respectively. The dynamic TSF (ratio) definition is same as defined before. The signals, SNR and throughput expressions seen at the respective PR and the AR nodes are similar to the ones defined in the previous sections. Note that the PR and AR devices in the proposed HAP scheme may be co-located or not. However, for analytic convenience related to the dynamic TSF design, we assume a co-located hybrid PR and AR set-up.

Over the second hop, the PR device uses the coefficient $\sqrt{\eta }$ to reflect the main transmit signal while the AR employs the AF-protocol to boost the transmit signal with the help of *w* and re-transmit the same to the end-user over the second hop. For the remainder of TP, i.e., (1−δ)T, the re-directed signals from the PR and AR experiences channel coefficients of $h_{2,P}$ and $h_{2,R}$, respectively. The corresponding signals, SNR and throughput expressions are similar as defined before. The overall effective throughput and the harvested energy expressions achieved at the end-user, incorporating the PS and TS schemes are respectively given by
$$R_{U,PR}=\left\{\begin{array}{c} R_{U,PR}^{\mathrm{PS}}=R_{1,D}^{\mathrm{PS}}+R_{2,P}^{\mathrm{PS}}+R_{2,R}^{\mathrm{PS}}\\ R_{U,PR}^{\mathrm{TS}}=R_{1,D}^{\mathrm{TS}}+R_{2,P}^{\mathrm{TS}}+R_{2,R}^{\mathrm{TS}} \end{array}\right.,$$
(13)$$E_{U,PR}=\left\{\begin{array}{c} E_{U,PR}^{\mathrm{PS}}=E_{1,D}^{\mathrm{PS}}\left(E[\right|y_{1,D}{|}^{2}],\rho )\\ \;+\left(1-\delta \right)E_{2,P}^{\mathrm{PS}}\left(E[\right|y_{2,P}{|}^{2}],\rho )\\ \;+\left(1-\delta \right)E_{2,R}^{\mathrm{PS}}\left(E[\right|y_{2,R}{|}^{2}],\rho )\\ E_{U,PR}^{\mathrm{TS}}=E_{1,D}^{\mathrm{TS}}\left(E[\right|y_{1,D}{|}^{2}],\tau )\\ \;+\left(1-\delta \right)E_{2,P}^{\mathrm{TS}}\left(E[\right|y_{2,P}{|}^{2}],\tau )\\ \;+\left(1-\delta \right)E_{2,R}^{\mathrm{TS}}\left(E[\right|y_{2,R}{|}^{2}],\tau ) \end{array}\right..$$

Considering a large number of parameters introduced in this work, we have provided the list of parameters and operators in Table 2. Additionally, the list of Greek symbols and their corresponding definitions are summarized in Table 3.

## 4. Problem Formulation and Solution

Herein, we present the problem to jointly optimize the data throughput, harvested energy and transmit power, subjected to QoS constraints for each considered schemes (viz., PGN, AGN, and HAP). Before proceeding to the main hypothesis, we introduce some variables to simplify the problem representation.

### 4.1. Variable Definitions to Assist the Problem Formulation

In this section, we define some additional parameters corresponding to the overall transmit power, data throughput and harvested energy expressions obtained at the end-user, the dynamic TSF ratios, the relay amplification factor and the TS and PS ratios; to refer to the aforementioned frameworks of PGN, AGN, and HAP altogether. In this context, the variables are sequentially defined in Table 4.

We additionally consider another parameter to assist the PS and TS ratio metrics, defined as follows.
(14)θ:={ρ:ForPSScheme;τ:ForTSScheme}.

It is clear that the newly defined parameters will adopt values according to the type of scenario chosen. Thus, we mathematically formulate the optimization problem in the following while incorporating the parameters defined in Table 4 and (Equation 14

### 4.2. Optimization Problem with Weighted Utility Function

We formulate the problem with weighted utility function to jointly optimize the data throughput, harvested energy and total transmit power of the source, while ensuring that the demanded throughput and the harvested power at the destination node are both individually above the given thresholds. The overall optimization problem incorporating the PGN, AGN and HAP systems is analytically represented as
(15)$$\begin{array}{ccc} \left(P1\right):\max_{P_{T},\delta ,\theta ,w} & & \frac{\omega_{R}R_{U}+\omega_{E}E_{U}}{\omega_{P}P_{T}} \end{array}$$(16)$$\begin{array}{ccc} \mathrm{subject}\;\mathrm{to}: & & \left(C1\right)\;:R_{1}\geq R_{2},\;\left(C2\right)\;:R_{U}\geq \varsigma ,\; \end{array}$$(17)$$\begin{array}{ccc} & & \left(C3\right)\;:E_{U}\geq \xi ,\;\left(C4\right)\;:0\leq \delta \leq 1,\; \end{array}$$(18)$$\begin{array}{ccc} & & \left(C5\right)\;:0\leq \theta \leq 1,\;\left(C6\right)\;:0<P\leq P^{⋆},\; \end{array}$$(19)$$\begin{array}{ccc} & & {\left(C7\right)}^{*}:0\leq {\left|w\right|}^{2}\leq P_{R},\; \end{array}$$
where ς and ξ are respectively the minimum demanded throughput and harvested energy by the destination SWIPT-capable user, $\omega_{R}$, $\omega_{E}$ and $\omega_{P}$ are the constant weighing factors corresponding to the data throughput, harvested energy and the transmit power, respectively. We use θ to interchangeably refer to the PS or TS splitting factor ρ or τ, respectively. In order to ensure feasibility during the second hop transmission, we make use of (C1) which essentially helps in designing δ and is also an indicator metric used at the transmitter for better resource management. It is noteworthy that (C1) has a completely different meaning and purpose in this context and is not to be confused with a similar constraint used in the case of regenerative relaying (DF). The constraints (C2) and (C3) are the minimum thresholds on the overall data rate and overall harvested energy, respectively. The upper and lower bounds of the dynamic TSF fraction and the SWIPT splitting factors are represented in (C4) and (C5) respectively. The constraint with maximum transmit power budget of the system is incorporated using (C6). Note that the variable *w* and ${\left(C7\right)}^{*}$ are considered only during the analysis of AGN and HAP systems. Focusing on the objective function, it is important to note the aspect of rate-energy (R-E) trade-off in SWIPT systems [34], which translates to either increase in data throughput at low EH demands, or vice-versa. Following this trade-off, the optimizer would intend to seek a balance point between the two metrics, viz., data throughput and harvested energy, such that both the metrics are maximized according to the stringent demands indicated in the constraints. In the following, we analyse (P1) and seek a suitable solution.

### 4.3. Proposed Solutions to the Above-Mentioned Problem

In this section, we present a couple of approaches to solve the problem in (P1). First, we employ a method based on the Lagrange dual approach to seek a close to optimal solution, without taking the computational complexity into consideration. The second technique is based on the alternating parameter based optimization, assisted by the Dinkelbach method. The nature of the latter process opens up the possibility of obtaining a sub-optimal solution within polynomial time. In this regard, the proposed solutions are discussed below.

#### 4.3.1. Method to Seek an Asymptotically Optimal Solution

The problem (P1) is difficult to solve since it has a non-convex form for both TS and PS schemes. However, it is clear from our optimization formulation that the dual method may be employed to seek an asymptotically optimal solution. Therefore, in order to obtain an analytical solution for the joint objective optimization problem in (P1), we assume D as the set of all possibilities of $P_{T},\delta ,\theta \;\mathrm{and}\;w$ which satisfy the constraints (*C*4)–(*C*7). Then the corresponding Lagrange dual function is written as(20)$$J\left(\Lambda \right)≜\max_{\{P_{T},\delta ,\theta ,w\}\in D}L\left(P_{T},\delta ,\theta ,w;\Lambda \right),$$where the Lagrangian is expressed as$$L\left(P_{T},\delta ,\theta ,w;\Lambda \right)=F\left(P_{T},\delta ,\theta ,w\right)-\lambda_{1}G\left(P_{T},\delta ,\theta ,w\right)\;$$(21)$$-\lambda_{2}H\left(P_{T},\delta ,\theta ,w\right)-\lambda_{3}I\left(P_{T},\delta ,\theta ,w\right),$$with $\Lambda =(\lambda_{1},\lambda_{2},\lambda_{3})⪰0$ denoting the vector of the dual variables associated with the constraints (*C*1)–(*C*3), respectively, while the related functions are as defined in Table 5. The dual optimization problem is hence represented as(22)$$\begin{array}{ccc} \left(P2\right):\min_{\Lambda } & & J(\Lambda ) \end{array}$$(23)$$\begin{array}{ccc} \mathrm{subject}\;\mathrm{to}: & & \Lambda ⪰0. \end{array}$$

Since it is explicit that a dual problem is always convex by definition [22], the gradient or subgradient-based methods can be used to minimize J(Λ) with guaranteed convergence. In this regard, an asymptotic technique based on the block-coordinate descent approach may be employed [27]. Herein, an alternating optimization of the involved parameters is performed in an iterative manner, until the convergence of the weighted utility function value. The intermediary closed-form expressions for the involved parameters in the stage-based optimization may be obtained using the Karush-Kuhn-Tucker (KKT) conditions [22]. Even though a solution could be obtained within polynomial time, the computation process of the proposed method is still complex and cumbersome, as inferred from the analysis in [27]. Therefore, we omit the corresponding derivations for brevity, and seek an even simpler method to solve (P1) as discussed in the succeeding section.

#### 4.3.2. Dinkelback Method Assisted Convex Programming

As discussed before, the problem (P1) involves a non-linear fractional programming, which is non-convex and difficult to solve directly. In such instances of non-convex non-linear fractional optimization, Dinkelbach method comes in handy to address this challenge [62]. Therefore, we transform the fractional form objective function into a difference of numerator and quasi-denominator form, using the following proposition.

**Proposition** **1.**
*The maximum achievable objective can be obtained using the following transformation, provided that*

$$\begin{array}{ccc} \max_{P_{T},\delta ,\theta ,w,\kappa^{⋆}} & & F_{N}\left(P_{T},\delta ,\theta ,w\right)-\kappa^{⋆}F_{D}\left(P_{T},\delta ,\theta ,w\right) \end{array}$$


(24)
$$\begin{array}{c} =F_{N}\left(P_{T}^{⋆},\delta^{⋆},\theta^{⋆},w^{⋆}\right)-\kappa^{⋆}F_{D}\left(P_{T}^{⋆},\delta^{⋆},\theta^{⋆},w^{⋆}\right)=0, \end{array}$$

*for $F_{N}\left(P_{T},\delta ,\theta ,w\right)\geq 0$ and $F_{D}\left(P_{T},\delta ,\theta ,w\right)\geq 0$, where*

(25)
$$\begin{array}{c} F_{N}\left(P_{T},\delta ,\theta ,w\right)=\omega_{R}R_{U}\left(P_{T},\delta ,\theta ,w\right)+\omega_{E}E_{U}\left(P_{T},\delta ,\theta ,w\right), \end{array}$$


(26)
$$\begin{array}{c} F_{D}\left(P_{T},\delta ,\theta ,w\right)=\omega_{P}P_{T}, \end{array}$$

*and*

(27)
$$\begin{array}{c} \kappa^{⋆}=\frac{F_{N}\left(P_{T}^{⋆},\delta^{⋆},\theta^{⋆},w^{⋆}\right)}{F_{D}\left(P_{T}^{⋆},\delta^{⋆},\theta^{⋆},w^{⋆}\right)}. \end{array}$$



**Proof.** Please refer to [62] for a proof of Proposition 1.    □

It is clear that the adequate and compulsory conditions are provided by Proposition 1 for developing suitable strategy to optimize the intended parameters. Specifically, it should be possible to transform the primal problem in (P1) having a fractional form-objective function into an equivalent optimization problem with a difference of numerator and quasi-denominator form-objective function (e.g., $F_{N}\left(P_{T},\delta ,\theta ,w\right)-\kappa^{⋆}F_{D}\left(P_{T},\delta ,\theta ,w\right)$). Consequently, the solutions for both the primal and transformed problems will be equivalent [28]. 

Herein, it is noteworthy that an optimal solution could be obtained if the equality in (Equation 24) holds [62]. The equality condition may then be correspondingly used to validate the optimality of the solution. Therefore, instead of directly addressing the problem with a fractional-form objective, we transform it to another equivalent optimization problem with subtractive-form objective function whilst meeting the conditions in Proposition 1, whose solution is summarized in Algorithm 1. 

In order to understand the working principle of Algorithm 1, we first note that the pivotal stage for the proposed Dinkelbach method-based solution is to develop an intermediate parameters’ optimization policy in order to solve the following fixed κ optimization problem (steps 2–6 in Algorithm 1,(28)$$\begin{array}{ccc} \left(P3\right):\max_{P_{T},\delta ,\theta ,w} & & \left(\omega_{R}R_{U}+\omega_{E}E_{U}\right)-\kappa^{⋆}\left(\omega_{P}P_{T}\right) \end{array}$$
(29)$$\begin{array}{ccc} \mathrm{subject}\;\mathrm{to}: & & \left(C1\right):R_{1}\geq R_{2},\;\left(C2\right):R_{U}\geq \varsigma ,\; \end{array}$$
(30)$$\begin{array}{ccc} & & \left(C3\right):E_{U}\geq \xi ,\;\left(C4\right):0\leq \delta \leq 1,\; \end{array}$$
(31)$$\begin{array}{ccc} & & \left(C5\right):0\leq \theta \leq 1,\;\left(C6\right):0<P\leq P^{⋆},\; \end{array}$$
(32)$$\begin{array}{ccc} & & {\left(C7\right)}^{*}:0\leq {\left|w\right|}^{2}\leq P_{R}.\; \end{array}$$

**Algorithm 1** Dinkelbach-assisted Alternating Parameter Optimization
1:**Initialize**: κ=0, any feasible δ,θ,w, and $\bar{\epsilon }$ : Threshold limit;2:**REPEAT** (for a given κ, iteration: n)3:  Solve (Equation 28)–(32) to obtain $P_{T}$, for given δ,θ,w;4:  Utilizing $P_{T}$, Solve (Equation 28)–(32) to obtain δ, for given θ,w;5:  Using $P_{T}$ and δ , Solve (Equation 28)–(32) to obtain θ, for given *w*;6:  Finally compute *w* by solving (Equation 28)–(32) via $P_{T}$, δ, θ;7: **IF** ($\omega_{R}R_{U}+\omega_{E}E_{U})-\kappa \left(\omega_{P}P_{T}\right)$≤$\bar{\epsilon }$8:  Convergence = **TRUE**;9:  **RETURN** $\left\{P_{T}^{⋆},\delta^{⋆},\theta^{⋆},w^{⋆}\right\}=\left\{P_{T},\delta ,\theta ,w\right\}$, $\kappa^{⋆}=\frac{\left(\omega_{R}R_{U}+\omega_{E}E_{U}\right)}{\omega_{P}P_{T}}$;10: **ELSE**11:   Set $\kappa =\frac{\left(\omega_{R}R_{U}+\omega_{E}E_{U}\right)}{\omega_{P}P_{T}}$ and n=n+1;12:   Convergence = **FALSE**;13: **END IF**14:**UNTIL** Convergence = **TRUE**.


Since the persisting non-convexity of problem in (P3) causes a hindrance in seeking an optimal solution directly via joint optimization of the intended variable, it is natural to explore other relevant possibilities for obtaining a suitable solution. In this vein, we note that the problem can be made more tractable by optimizing a single parameter while keeping the other parameters fixed since the convexity of the problem is more probable. Such an alternating parameter based optimization technique has been found effective for solving various complicated optimization problems [63,64]. Herein, the optimality of the solution is, however, compromised. Next, we observe that following an alternating parameter based technique, each corresponding sub-problem belong to the category of disciplined convex-concave programming (DCCP) [65]. The consistently occurring convex-concave nature in (C1) essentially contributes towards making (P3) a DCCP. Therefore, suitable solutions could be obtained using standard optimization methods, e.g., the DCCP package in CVXPY [66]. Correspondingly, the problem (Equation 28)–(Equation 32) can be solved using the proposed Dinkelbach-assisted Alternating Parameter Optimization, summarized in Algorithm 1.

Regarding the convergence of the proposed Dinkelbach-assisted Alternating Parameter Optimization based solution, we initially prove that the auxiliary variable κ appreciates in each iteration. Following this, we prove that the value of κ converges to the optimal $\kappa^{⋆}$ for sufficiently large number of iterations, such that the optimality condition in Proposition 1 is satisfied. More specifically, $F_{N}\left(P_{T},\delta ,\theta ,w\right)-\kappa^{⋆}F_{D}\left(P_{T},\delta ,\theta ,w\right)=\hat{F}\left(\kappa^{⋆}\right)=0$. Let us consider $\left\{P_{T}^{⋆},\delta^{⋆},\theta^{⋆},w^{⋆}\right\}$ as the optimized policies in the nth iteration. Suppose $\kappa \left(n\right)\neq \kappa^{⋆}$ and $\kappa \left(n+1\right)\neq \kappa^{⋆}$ represent the relevant values in the nth and (n+1)th iterations, respectively. We note from [62] that $\hat{F}\left(\kappa \left(n\right)\right)>0$ and $\hat{F}\left(\kappa \left(n+1\right)\right)>0$ hold. On the other hand, in the proposed algorithm, $\kappa \left(n+1\right)=\frac{F_{N}\left(P_{T},\delta ,\theta ,w\right)}{F_{D}\left(P_{T},\delta ,\theta ,w\right)}$. Thus, we can express κ(n) as
(33)$$\begin{array}{c} \hat{F}\left(\kappa \left(n+1\right)\right)=F_{N}\left(P_{T},\delta ,\theta ,w\right)-\kappa \left(n\right)F_{D}\left(P_{T},\delta ,\theta ,w\right)\\ =F_{D}\left(P_{T},\delta ,\theta ,w\right)\left(\kappa (n+1)-\kappa (n)\right).\; \end{array}$$

Since $F_{D}\left(P_{T},\delta ,\theta ,w\right)=\omega_{P}P_{T}>0$, we have κ(n+1)>κ(n). Therefore, we can show that as long as the number of iterations is large enough, $\hat{F}\left(\kappa \left(n\right)\right)$ will eventually approach zero and satisfy the optimality condition as stated in Proposition 1.

## 5. Numerical Results

In this section, we present a comparative study among the HAP, AGN and PGN systems. The three cases are analyzed using the MATLAB R2019b, with optimization performed via fmincon(·) solver present in the optimization toolbox [67], where the solutions are obtained with the help of interior point algorithm [22].

### 5.1. Simulation Set-Up

We assume an ITU-R outdoor framework (site-general model for propagation within street canyons) [68] to generate channel realizations with the path-loss exponent:
(34)$$PL\left(d,f\right)=10a\log_{10}\left(d\right)+b+10c\log_{10}\left(f\right)+N\left(0,\sigma \right)\;\mathrm{dB},$$where *d* is distance between the transmitting and receiving stations (m), *f* is the operating frequency (GHz), the coefficients *a*, *b*, and *c* are associated with the increase of the path loss with distance, the offset value of the path loss, and the increase of the path loss with frequency, respectively, and N(0,σ) is a zero mean Gaussian random variable with a standard deviation σ (dB). The channels coefficients for the direct link and the dual-hop links are generated accordingly. Specifically, we choose: *f* = 24 GHz, *d* is randomly chosen between 8 m and 10 m in case of PR/AR device and randomly between 12 m and 15 m for the end-user with respect to the transmitter [69], *a* = 2.12, *b* = 29.2, *c* = 2.11 and σ = 5.06 dB. The simulation results presented in this section assume an overall bandwidth of B=50 MHz. We set $\sigma_{n_{B}}^{2}$ = $\sigma_{n_{R}}^{2}$ = $\sigma_{n_{u_{a}}}^{2}$ = −150 dBW/Hz, and $\sigma_{n_{u_{ID}}}^{2}$ = −110 dBW/Hz. The PR’s reflection efficiency coefficient is chosen as η = 0.78 (corresponding to 1.1 dB loss [70]). The constants corresponding to the non-linear EH circuit are chosen as $E^{\prime }$ = 2.8 mJ, α = 1500, and β = 0.0022 [27,58]. The harvested energy demand is assumed to be ξ = −70 dBW, and the weighing factors are chosen as $\omega_{R}$ = 1 units/Gbit, $\omega_{E}$ = 1 units/nJ, and $\omega_{P}$ = 1 units/W. We use two test cases for the data demands of ς = 1.00 Gbits and ς = 1.05 Gbits. As a benchmark scheme, we assume equal time distribution for both the hops in all the cases, i.e., fixed δ = 0.5. Furthermore, an average of 1000 random channel realizations is presented for each experiment.

### 5.2. Experimental Findings and Analysis

In Figure 3a,b, we present the performance analyses of HAP, AGN and PGN in terms of the weighted utility function for the PS and TS schemes, respectively, versus the maximum limitation on the total transmit power. We observe that all the considered schemes can provide appreciable gains over their respective benchmark methods. The weighted utility function values for both PS and TS increases with growing overall maximum transmit power budget ($P^{⋆}$) of the system. On the other hand, the performance is adversely affected when the data demand is increased from ς = 1.00 Gbits to ς = 1.05 Gbits, which is due to the joint optimization in the weighted utility function. Particularly, this effect is due to the well-known rate-energy trade-off incurred because of the joint maximization of data and harvested energy along with the minimization of total transmit power. In the case of PS, we notice improvements of 10% and 12.5% for HAP over the AGN and PGN scenarios, respectively for the data demand of ς = 1.00 Gbits, and improvements of 2% and 4% respectively for the data demand of ς = 1.05 Gbits, with an average decrement of 53% for a transition from former to the latter demand. Concerning the TS, we observe gains of 25% and 28% for HAP over the AGN and PGN scenarios, respectively for the data demand of ς = 1.00 Gbits, and improvements of 11.5% and 12.5%, respectively for the data demand of ς = 1.05 Gbits, with an average overall decrement of 80.5% for a transition from ς = 1.00 Gbits to ς = 1.05 Gbits. Clear benefits for HAP are observed over the AGN and PGN counterparts, with PS scheme outperforming the TS.


We now investigate the system performance from the perspective of so-called R-E trade-off in SWIPT systems [34]. Correspondingly, we show in Figure 4 the effect on harvested energy with varying data demand. It is noteworthy that the considered analysis may be easily carried out by keeping $\omega_{R}=0$ in (P1). In this vein, it is also important to mention that besides the use of $\omega_{R}$ and $\omega_{E}$ as the cost functions for data and energy metrics, respectively; their binary forms may be utilized to study the corresponding entities independently. Herein, we observe the (expected) decreasing nature of the harvested energy with increasing data demands for all the cases. The HAP system is found to outperform the AGN and PGN as anticipated, whereas the benefits of employing dynamic TSF are clearly seen over the fixed TSF (δ). We additionally note that the performance of the PS mechanism is way superior than the TS counterpart.

In Figure 5a,b, we show the effects on the weighted utility functions corresponding to the PS and TS, respectively, with varying TSF (i.e., fixed δ at an instance). Intuitively, when the TSF is small, the second hop reflection/re-transmission is expected to happen for significantly large amount of time with the considered TP. This would naturally enable better collection and interpretation of the energy and data, respectively, at the end-user following the signal synchronization process (as assumed). In other case, when the TSF is large, the second hop reflection/re-transmission would suffer due to a highly compact TP, which affects the overall throughput and hence the output at the end-user. We note from Figure 5a,b that the above-mentioned intuitive discussion is validated, wherein the depreciating values of the weighted utility function are observed with the increasing values of
δ. On the other hand, the HAP system is found to outperform the AGN and PGN counterparts with PS providing significant gains over the TS scheme, thereby following in-line with the previously reported results.

In order to allow fair comparison among the different analyzed approaches, we calculate confidence intervals for the results and present the corresponding error bars in Figure 6a,b, pertaining to the PS and TS schemes, respectively. To compute the confidence interval, we use the formula:
$\chi \pm \gamma \frac{\sigma }{\sqrt{\nu }}$, where χ is the mean, γ is the Z-value from the Table 6, σ is the standard deviation, and ν is the number of observations [71]. To proceed, we consider ν=1000 observations and compute the weighted utility functions corresponding to the PS and TS schemes for different values of the maximum transmit power budget ($P^{⋆}$) and keeping ς=1.00 Gb. Therefore, it is clear that for 1000 instances the weighted utility functions are calculated per single consideration of $P^{⋆}$. Correspondingly, the values of χ and σ can be computed. For our experiment, we assume γ=1.960 pertaining to a confidence level of 90% to obtain the error bars shown in Figure 6a,b, for the PS and TS schemes, respectively. The HAP systems in both the cases are seen to outperform the AGN and PGN systems for PS and TS schemes. The fairness measure in terms of error bars is also explicit in this regard.

### 5.3. General Outcomes and Trailing Discussion

From the results reported in the previous section and following an intuitive analysis, one important obvious observation is the superior performance of the HAP over the AGN and PGN systems. The significant benefits observed via the PS scheme over the TS are also explicit. The proposed framework shows that the existing AGN and PGN type of standalone systems may be fused together leading to the possibility of reaping maximum benefits within the system. In contrast, the RIS systems have evolved as the promising solutions, however, there are still considerable number of challenges involved in relation to their practical deployment. In this context, the primary concerns revolve around the ways to achieve flexible reflections and powering/switching mechanisms. Additionally, the practical implementation of dynamic TSF-enabled relaying may be challenging to achieve in practice [72], however, we cannot completely rule out this possibility on this type of AF-based cooperative system deployment. Alternatively, the benefits of equal-TSF based systems have also been addressed herein, which might be easier to deploy from the practical viewpoint. Nonetheless, it is needless to mention that an intermediary possibility of HAP comes in handy where an alternative deployment to RIS may be put into practice while the relevant active components/mechanisms are under development. The proposed methodology does not only seem to be a cost and energy efficient approach, but also serves as a promising solution for harnessing the maximum gains from the transmit signal power (within the considered wireless communications system).

## 6. Conclusions

In this invited paper, we investigated a hybrid active-and-passive (HAP) relaying scheme to not only improve the network coverage, but also to facilitate SWIPT in a 5G and beyond/6G Greencom environment, where the end-users are capable of decoding information and harvesting energy simultaneously, according to the PS and TS schemes while using the non-linear EH device for the latter. We formulated an optimization problem to jointly address the respective demands of data throughput, harvested energy and total transmit power, under some QoS constraints. In this context, we provided several polynomial computation-time based solutions which employ the Lagrange dual and Dinkelbach method assisted convex programming, respectively. We showed significant benefits of adopting HAP over the AGN and PGN counterpart systems with the help of numerical results, where the PS scheme was found to outperform the TS concerning SWIPT at the end-user. Considering significant benefits of the proposed HAP system, this work may be extended into several interesting directions in the future. Some extension possibilities include the investigation with distinct placement of the reflection and re-transmission elements, the incorporation of multiple antennas at the concerned devices, and the consideration of multiple HAP nodes and multiple users whose demands are addressed with the help of a multi-carrier framework. The examination of a cooperative system with DF protocol-enabled relays instead of AF (as considered herein) is also another interesting research prospect. Furthermore, the possibility of precoding/beamforming could also be considered to enhance the performance of the multiple antenna-enabled HAP systems.

## Figures and Tables

**Figure 1 sensors-21-06013-f001:**
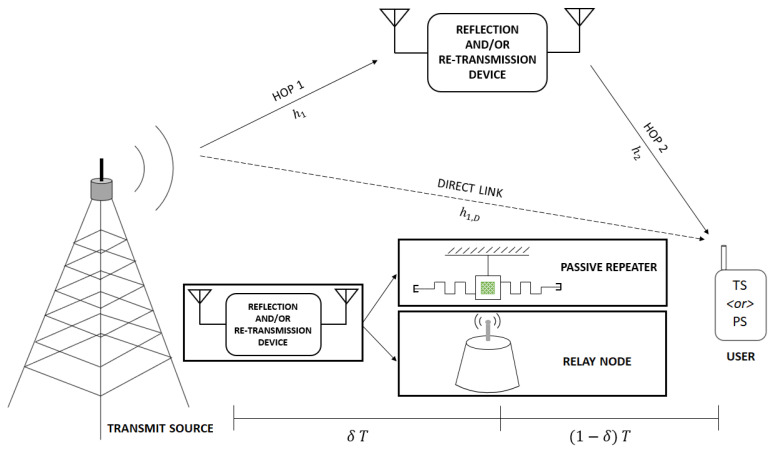
Dual hop system with a PR and/or AR device.

**Figure 2 sensors-21-06013-f002:**
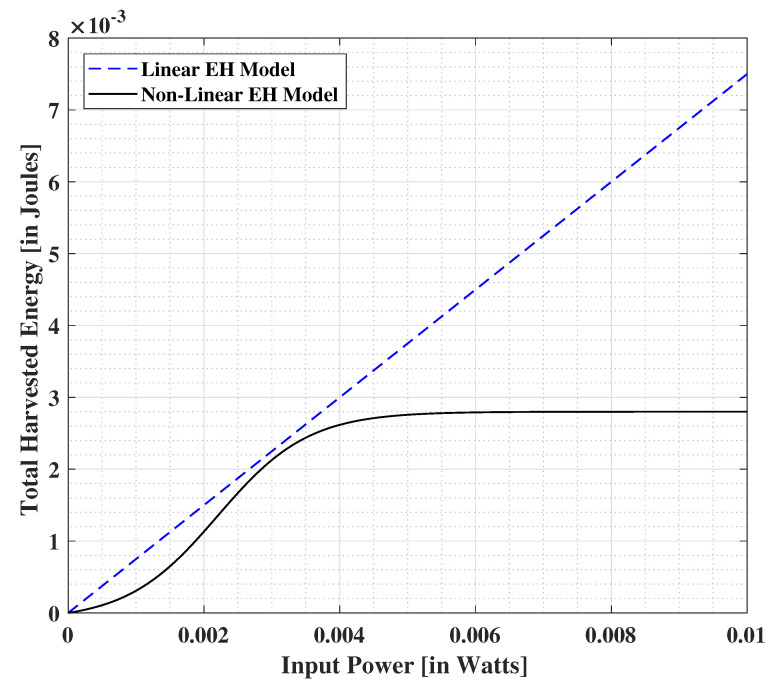
Comparison between the energy extraction capabilities of linear and non-linear energy harvesting models.

**Figure 3 sensors-21-06013-f003:**
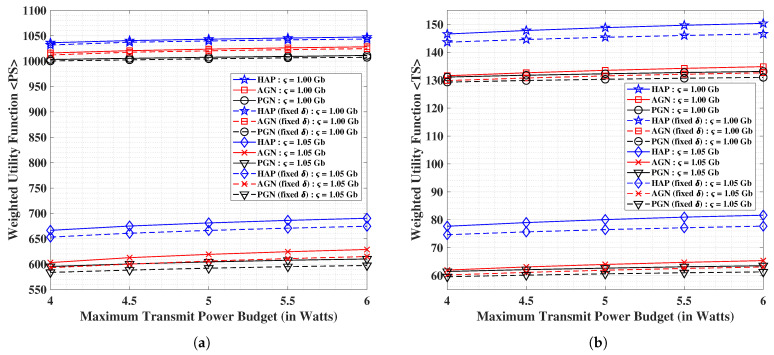
Performance analysis of the weighted utility function for (**a**) PS scheme, and (**b**) TS scheme; versus the maximum limitation on the total transmit power.

**Figure 4 sensors-21-06013-f004:**
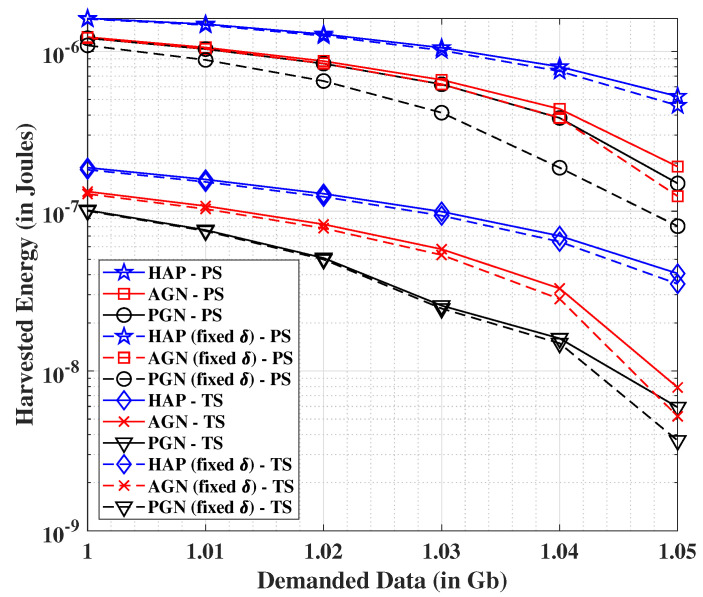
Analysis of harvested energy for PS and TS schemes versus the demanded data (R-E trade-off).

**Figure 5 sensors-21-06013-f005:**
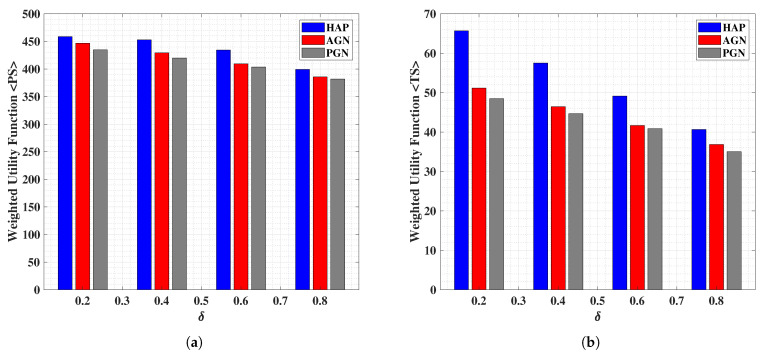
Performance analysis of the weighted utility function for (**a**) PS scheme, and (**b**) TS scheme; versus the variation in the (fixed) TSF δ, while comparing HAP, AGN and PGN

**Figure 6 sensors-21-06013-f006:**
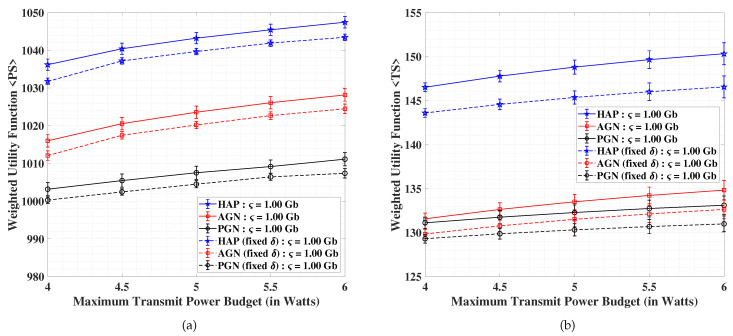
Analysis of harvested energy for (**a**) PS scheme, and (**b**) TS scheme; versus the demanded data (R-E trade-off) with error bars pertaining to a confidence level of 95%.

**Table 1 sensors-21-06013-t001:** List of Abbreviations.

Acronym	Full-Form
3GPP	3rd Generation Partnership Project
5G	Fifth Generation Mobile Wireless Communications
6G	Sixth Generation Mobile Wireless Communications
AF	Amplify-and-Forward
AGN	Active Relay Device-based Green Communication Network
AR	Active Relay
AWGN	Additive White Gaussian Noise
CSI	Channel State Information
DCCP	Disciplined Convex-Concave Programming
DF	Decode-and-Forward
EH	Energy Harvesting
FR1	Frequency Range-1 for 5G
FR2	Frequency Range-2 for 5G
Greencom	Green Communications
HAP	Hybrid Active-and-Passive
ID	Information Decoding
IoT	Internet-of-Things
IRS	Intelligent Reflecting Surface
KKT	Karush-Kuhn-Tucker
MRC	Maximum Ratio Combining
NR	New Radio
PGN	Passive Repeater Device-based Green Communication Network
PR	Passive Repeater
PS	Power Splitting
QoS	Quality-of-Service
RIS	Reconfigurable Intelligent Surface
SNR	Signal-to-Noise-Ratio
SWIPT	Simultaneous Wireless Information and Power Transfer
TP	Time Period
TS	Time Switching
TSF	Time-Splitting Factor

**Table 2 sensors-21-06013-t002:** List of Parameters and Operators—Definitions.

Notation	Definition
*a*	Coefficient associated with increase of path loss with distance
B	Overall bandwidth
*b*	Coefficient associated with the offset value of path-loss
*c*	Coefficient associated with increase of path loss with frequency
*d*	Distance between the transmitting and receiving stations (in meters)
$E^{\prime }$	Maximum harvested energy obtained on the saturation of the EH circuit
E(x)	Non-linear EH (sigmoidal) expression with input power *x*
$E^{\mathrm{PS}}\left(x,\rho \right)$	Non-linear EH expression for PS scheme
$E^{\mathrm{TS}}\left(x,\tau \right)$	Non-linear EH expression for TS scheme
$E_{U}^{\mathrm{PS}}$	Harvested energy at the end-user with PS scheme in the PGN/AGN scenario
$E_{U}^{\mathrm{TS}}$	Harvested energy at the end-user with TS scheme in the PGN/AGN scenario
$E_{U}$	Parameter to refer to $E_{U}^{\mathrm{PS}}$ or $E_{U}^{\mathrm{TS}}$ according to the chosen scheme
$E_{U,PR}^{\mathrm{PS}}$	Overall harvested energy at the end-user with PS scheme in the HAP scenario
$E_{U,PR}^{\mathrm{TS}}$	Overall harvested energy at the end-user with TS scheme in the HAP scenario
$E_{U,PR}$	Parameter to refer to $E_{U,PR}^{\mathrm{PS}}$ or $E_{U,PR}^{\mathrm{TS}}$ according to the chosen scheme
E[·]	Expectation value operator
*f*	Operational frequency of the system
$h_{1,D}$	Channel coefficient for the direct link between transmit source and end-user
$h_{1,P}$	Channel coefficient for the first phase of indirect link pertaining to PR device
$h_{1,R}$	Channel coefficient for the first phase of indirect link pertaining to AR device
$h_{1}$	Parameter to refer to $h_{1,P}$ or $h_{1,R}$ according to the chosen scheme
$h_{2,P}$	Channel coefficient for the second phase of indirect link pertaining to PR device
$h_{2,R}$	Channel coefficient for the second phase of indirect link pertaining to AR device
$h_{2}$	Parameter to refer to $h_{2,P}$ or $h_{2,R}$ according to the chosen scheme
J(·)	Lagrange Dual Function
L(·)	Lagrangian function operator
n${}_{P}$	AWGN introduced at the PR device
n${}_{R}$	AWGN introduced at the AR device
$n_{1}$	Parameter to refer to $n_{P}$ or $n_{R}$ according to the chosen scheme
$n_{u_{a}}$	AWGN introduced due to the antenna element of the end-user
$n_{u_{ID}}$	The noise introduced by the baseband processing circuit at the end-user
$P_{T}$	Transmit power at the source
$P_{R}$	Maximum overall power available at the relay
$P^{⋆}$	Overall power limitation for the transmitter-relay system
$R_{U}^{\mathrm{PS}}$	Data throughput at the end-user with PS scheme in the PGN/AGN scenario
$R_{U}^{\mathrm{TS}}$	Data throughput at the end-user with TS scheme in the PGN/AGN scenario
$R_{U}$	Parameter to refer to $R_{U}^{\mathrm{PS}}$ or $R_{U}^{\mathrm{TS}}$ according to the chosen scheme
$R_{U,PR}^{\mathrm{PS}}$	Overall data throughput at the end-user with PS scheme in the HAP scenario
$R_{U,PR}^{\mathrm{TS}}$	Overall data throughput at the end-user with TS scheme in the HAP scenario
$R_{U,PR}$	Parameter to refer to $R_{U,PR}^{\mathrm{PS}}$ or $R_{U,PR}^{\mathrm{TS}}$ according to the chosen scheme
$R_{1,D}^{\mathrm{PS}}$	Data throughput at the end-user with PS scheme via direct link
$R_{1,D}^{\mathrm{TS}}$	Data throughput at the end-user with TS scheme via direct link
$R_{1}$	Parameter to refer to $R_{1,D}^{\mathrm{PS}}$ or $R_{1,D}^{\mathrm{TS}}$ according to the chosen scheme
$r_{1,P}$	Transmit signal seen at the PR device
$r_{1,R}$	Transmit signal seen at the AR device
$r_{1}$	Parameter to refer to $r_{1,P}$ or $r_{1,R}$ according to the chosen scheme
*s*	Symbol tranmsitted from the source
*T*	Time period
*w*	Complex amplification coefficient of the AR device
$y_{1,D}$	Signal received by the end-user via direct link
$y_{2,P}$	Signal received by the end-user via PR device over the second phase
$y_{2,R}$	Signal received by the end-user via AR device over the second phase
$y_{2}$	Parameter to refer to $y_{2,P}$ or $y_{2,R}$ according to the chosen scheme

**Table 3 sensors-21-06013-t003:** List of Greek Symbols and Corresponding Definitions.

Symbol	Definition
(α,β)	Constants corresponding to the capacitor and diode turn-on voltage at EH circuit
χ	Mean corresponding to the confidence interval formula
δ	Time splitting factor corresponding to TP
$\bar{\epsilon }$	Threshold limit corresponding to Algorithm 1
$\sqrt{\eta }$	Reflection efficiency coefficient of the PR device
γ	Z-value corresponding to the confidence interval formula
κ	Parameter to compute the intermediary fraction during the Dinkelbach process
Λ	Vector corresponding to the Lagrange dual variables: $\left(\lambda_{1},\lambda_{2},\lambda_{3}\right)$
ν	The number of observations corresponding to the confidence interval formula
$\omega_{E}$	Weighing coefficient corresponding to the harvested energy
$\omega_{P}$	Weighing coefficient corresponding to the transmit power
$\omega_{R}$	Weighing coefficient corresponding to the data throughput
ρ	Power splitting ratio
σ	Standard deviation
$\sigma_{n_{u_{a}}^{2}}$	Noise variance corresponding to $n_{u_{a}}$
$\sigma_{n_{u_{ID}}^{2}}$	Noise variance corresponding to $n_{u_{ID}}$
$\sigma_{n_{P}^{2}}$	Noise variance corresponding to $n_{P}$
$\sigma_{n_{R}^{2}}$	Noise variance corresponding to $n_{R}$
$\sigma_{n_{1}^{2}}$	Parameter to refer to $\sigma_{n_{P}^{2}}$ or $\sigma_{n_{R}^{2}}$ according to the chosen scheme
τ	Time-switching ratio
Θ	Metric to represent τ or ρ interchangeably
$\Upsilon_{1,D}^{\mathrm{PS}}$	SNR obtained at the end-user with PS scheme via direct link
$\Upsilon_{1,D}^{\mathrm{TS}}$	SNR obtained at the end-user with TS scheme via direct link
$\Upsilon_{1,P}$	SNR estimated at the PR via first hop indirect link
$\Upsilon_{1,R}$	SNR estimated at the AR via first hop indirect link
$\Upsilon_{1}$	Parameter to refer to $\Upsilon_{1,P}$ or $\Upsilon_{1,R}$ according to the chosen scheme
$\Upsilon_{2,P}^{\mathrm{PS}}$	SNR obtained at the end-user with PS scheme via second hop PR device link
$\Upsilon_{2,P}^{\mathrm{TS}}$	SNR obtained at the end-user with TS scheme via second hop PR device link
$\Upsilon_{2,R}^{\mathrm{PS}}$	SNR obtained at the end-user with PS scheme via second hop AR device link
$\Upsilon_{2,R}^{\mathrm{TS}}$	SNR obtained at the end-user with TS scheme via second hop AR device link
$\Upsilon_{2}^{\mathrm{PS}}$	Parameter to refer to $\Upsilon_{2,P}^{\mathrm{PS}}$ or $\Upsilon_{2,R}^{\mathrm{PS}}$ according to the chosen scheme
$\Upsilon_{2}^{\mathrm{TS}}$	Parameter to refer to $\Upsilon_{2,P}^{\mathrm{TS}}$ or $\Upsilon_{2,R}^{\mathrm{TS}}$ according to the chosen scheme
ς	Minimum demanded data throughput
ξ	Minimum demanded harvested energy
Ξ	Metric combining reflection/amplification coefficients of PR/AR, respectively

**Table 4 sensors-21-06013-t004:** Parameter definitions to assist problem formulation.

	Parameter	*P*	$R_{1}$	$R_{2}$	$R_{U}$	$E_{U}$
System						
PGN		$P_{T}$	$R_{1,P}$	$R_{2,P}$	$R_{U,P}$	$E_{U,P}$
AGN		$P_{T}+P_{R}$	$R_{1,R}$	$R_{2,R}$	$R_{U,R}$	$E_{U,R}$
HAP		$P_{T}+P_{R}$	$\left[\begin{array}{c} R_{1,P}\\ R_{1,R} \end{array}\right]$	$\left[\begin{array}{c} R_{2,P}\\ R_{2,R} \end{array}\right]$	$R_{U,PR}$	$E_{U,PR}$

**Table 5 sensors-21-06013-t005:** Definitions to comprise the Lagrangian expression.

$F(P_{T},\delta ,\theta ,w)$	$G(P_{T},\delta ,\theta ,w)$	$H(P_{T},\delta ,\theta ,w)$	$I(P_{T},\delta ,\theta ,w)$
$\frac{\omega_{R}R_{U}+\omega_{E}E_{U}}{\omega_{P}P_{T}}$	$R_{2}-R_{1}\leq 0$	$\varsigma -R_{U}\leq 0$	$\xi -E_{U}\leq 0$

**Table 6 sensors-21-06013-t006:** Determination of Z-value according to the percentage of confidence level.

Percentage	80%	85%	90%	95%	99%	99.5%	99.9%
**Z-value**	1.282	1.440	1.645	1.960	2.576	2.807	3.291

## Data Availability

Not applicable.

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
