# Peer review of "Hybrid Active-and-Passive Relaying Model for 6G-IoT Greencom Networks with SWIPT"

_sensors, 2021, doi:10.3390/s21186013_

Round 1

Reviewer 1 Report

  1. General remark

First of all, I would like to congratulate the Authors, this work presents interesting theoretical and numerical confirmation of the investigated problem. This paper proposes innovation in simultaneous wireless information and power transfer and responds perfectly to increasing demands of Internet of Things and future generations of wireless networks e.g. 5G and 6G.

The paper is well organized, there are 6 sections: (i) intro, (ii) system model; (iii) study of network scenarios; (iv) problem formulation; (v) numerical modelling; and (vi) conclusions.

The Authors are presenting in detailed way the used mathematical model and theory used in modelling. Formulas are clear and meanings of variables is explained.

The weakest point of this paper is lack of experimental implementation of proposed algorithms. But the Authors discuss about this deficiency explaining it by the difficulties in achieving flexible reflections and achieving switching/powering mechanisms. Despite that, this study is attractive and inspiring when going towards practical implementations.

I therefore recommend to the Editor to accept this paper under Minior revisions for publication in Sensors.

  1. Suggestions, remarks and doubts
    1. I highly recommend to prepare a list of used symbols and abbreviations. The paper is full of abbreviations and it is difficult for the reader to find fist appearance of the abbreviation in the text to check what is the full name.
    2. Line 65 – I think there are misplaced words there is “of thermal noise at the very RIS small elements” I think “RIS” should be after “small”.;
    3. Line 166 – double “the” in the sentence;

Reviewer 2 Report

This paper explores a novel hybrid active-and-passive relaying scheme with simultaneous wireless information and power transfer (SWIPT) for IoT. The manuscript is well written, and presents a solid mathematical/theoretical analysis. Following are some comments:

1. A Table of symbols with definitions is recommended for this paper.

2. In Line 153 it reads: "In order to estimate the amount of harvested energy at the end-user, we adopt the sigmoidal function based non-linear EH model." What theoretical considerations justify the selection and use of this particular EH model?

3. Make sure that every acronym is defined when used for the first time.

4. The 5G frequency range 2 starts from frequencies above 24 GHz. Also, distances in the order of 10 meters seem short compared with 5G range capabilities. Is there any theoretical justification or reference for the selection of these parameters for your simulations?

5. Your simulations seem to cover the scenario of one transmitter and a single user/receiver. How can your results and conclusions be extrapolated to an scenario of an actual network, with tens or hundreds of users/receivers in a given area?

Reviewer 3 Report

The manuscript proposes a hybrid relaying scheme to allow simultaneous wireless information and power transfer. The idea is exciting, and the results surpass those of a related technique. The manuscript is well-written. However, some points deserve more attention before the article is suitable for publication.

1) The authors should consider removing the term "invited paper" from the abstract.

2) The authors should include a background section on the topics necessary for a non-expert reader to understand the proposal. 

3) A section to discuss related work is necessary.

4) The relationship between the proposed solution and 6G networks is not clear. The authors must consider 6G-specific parameters in their simulation or adapt the paper's title if this simulation is not feasible.

5) The reason for selecting the simulation parameters must be discussed. 

6) The authors should calculate confidence intervals for the results and present error bars to allow fair comparison among the different analyzed approaches.  
